# Injectable Peptide Hydrogel Encapsulation of Mesenchymal Stem Cells Improved Viability, Stemness, Anti-Inflammatory Effects, and Early Stage Wound Healing

**DOI:** 10.3390/biom12091317

**Published:** 2022-09-17

**Authors:** Quan Li, Guangyan Qi, Dylan Lutter, Warren Beard, Camila R. S. Souza, Margaret A. Highland, Wei Wu, Ping Li, Yuanyuan Zhang, Anthony Atala, Xiuzhi Sun

**Affiliations:** 1Carl and Melinda Helwig Department of Biological and Agricultural Engineering, Kansas State University, Manhattan, KS 66506, USA; 2Department of Grain Science and Industry, Kansas State University, Manhattan, KS 66506, USA; 3Department of Clinical Sciences, College of Veterinary Medicine, Kansas State University, Manhattan, KS 66506, USA; 4Texas Specialty Veterinary Services, Boerne, TX 78006, USA; 5Wisconsin Veterinary Diagnostic Laboratory, University of Wisconsin-Madison, Madison, WI 53706, USA; 6Department of Chemistry, Kansas State University, Manhattan, KS 66506, USA; 7Wake Forest Institute Regenerative Medicine, Wake Forest University, Winston-Salem, NC 27151, USA

**Keywords:** hydrogel, cell delivery, mesenchymal stem cell, wound healing

## Abstract

Human-adipose-derived mesenchymal stem cells (hADMSCs) are adult stem cells and are relatively easy to access compared to other sources of mesenchymal stem cells (MSCs). They have shown immunomodulation properties as well as effects in improving tissue regeneration. To better stimulate and preserve the therapeutic properties of hADMSCs, biomaterials for cell delivery have been studied extensively. To date, hyaluronic acid (HA)-based materials have been most widely adopted by researchers around the world. PGmatrix is a new peptide-based hydrogel that has shown superior functional properties in 3D cell cultures. Here, we reported the in vitro and in vivo functional effects of PGmatrix on hADMSCs in comparison with HA and HA-based Hystem hydrogels. Our results showed that PGmatrix was far superior in maintaining hADMSC viability during prolonged incubation and stimulated expression of SSEA4 (stage-specific embryonic antigen-4) in hADMSCs. hADMSCs encapsulated in PGmatrix secreted more immune-responsive proteins than those in HA or Hystem, though similar VEGF-A and TGFβ1 release levels were observed in all three hydrogels. In vivo studies revealed that hADMSCs encapsulated with PGmatrix showed improved skin wound healing in diabetic-induced mice at an early stage, suggesting possible anti-inflammatory effects, though similar re-epithelialization and collagen density were observed among PGmatrix and HA or Hystem hydrogels by day 21.

## 1. Introduction

Mesenchymal stem cells (MSCs) are adult stem cells found in bone marrow, adipose tissue, synovial fluid, and other human tissues [1]. MSCs can generate and secrete a large variety of bioactive factors that can modulate immune response [2,3,4], promote angiogenesis [5], stimulate cell survival and proliferation, and affect differentiation of specific cells for tissue repair [6]. Studies have demonstrated great potential of MSCs and their secretome in alleviation or treatment of diseases, such as osteoarthritis [7,8,9,10,11], type 2 diabetes mellitus [12,13], obesity-induced metabolic dysregulation [14], cardiovascular diseases [5], and graft-versus-host disease (GVHD) [4]. Allogeneic MSC transplants were proven safe by multiple clinical studies [15,16,17], which expands therapeutic potentials of MSCs. MSCs can be isolated from a variety of tissue sources, including bone marrow, adipose tissue, and cord blood [18]. Human-adipose-derived MSCs (hADMSCs) are more easily accessible than MSCs from other sources, exhibit comparable therapeutic effects [18], and have been widely adopted for cell therapy research [8,9,14]. MSCs can be used in suspension directly after harvesting or after preconditioning methods that alter cell properties [4,19,20,21,22] or promote secretion of therapeutic factors [23,24,25,26]. Existing preconditioning methods commonly used are hypoxia [27,28,29], stimulation with inflammatory factors [30,31], and 3D configuration [32]. Various studies have investigated the formation of MSC aggregates that occur in a suspension setting and have found that such aggregate formation can negatively alter MSC functionalities, such as stemness and therapeutic factor secretion [19,32,33]. For this reason, researchers have focused on 3D hydrogel encapsulation, as it is not only relatively easy to operate, but also serves as a delivery system [34,35]. Shear-thinning hydrogels are usually preferred as they allow for injection, which simplifies transplantation and minimizes patient pain. In addition, hydrogel reduces mechanical stress on cells during injection, offers environmental cues to improve cell viability, and promotes secretion of therapeutic factors [36]. Both non-crosslinked hyaluronic acid (HA) [37] and hyaluronan-based hydrogel (Hystem) [38] have shown positive effects on MSCs and have been widely adopted by researchers as promising delivery materials in cell-based therapies [37,38,39,40]. However, the non-crosslinked HA gel and Hystem hydrogel are still relatively hard to handle, HA gel is sticky, while Hystem is temperature sensitive, which hinder their application in clinical settings. It is necessary to identify potential alternatives to advance clinical applications. 

In addition to HA-based hydrogels, peptide hydrogels have gained attention in recent years. Peptides can be designed from scratch to accommodate specific needs and are fully synthetic, which ensures consistency across batches. Rationally designed peptide hydrogels can self-assemble to form nanoweb structures, mimicking the natural ECM, thus, supporting cell performance in a 3D space [41,42,43,44]. Several peptide hydrogels have been utilized in life science research, including EAK16, RADA16 (commercial name PuraMatrix, BD Bioscience), Fmoc-FF, Fmoc-RGD, and h9e (commercial name PGmatrix, PepGel LLC). Among these peptide hydrogels, PGmatrix has shown significant advantages in 3D cell cultures [42,45,46,47] and in vivo virus delivery by injection [48,49]. PGmatrix possesses tunable nanoweb pore size, gel strength and indefinite sol–gel recovery properties [47], and superior biocompatibility, without causing inflammatory effects in subcutaneous injection [50]. We hypothesized that PGmatrix would provide improved functional cyto-properties of MSCs in 3D encapsulation, as well as in vivo delivery.

The present study was aimed to compare the in vitro and in vivo properties of human-adipose-derived mesenchymal stem cells (hADMSCs) in 3D encapsulation within non-crosslinked HA, Hystem, and PGmatrix hydrogels. PGmatrix-supported prolonged viability maintenance led to high expression of the stem cell marker stage-specific embryonic antigen-4 (SSEA4) and elevated release of immune-response-related factors in hADMSCs compared to HA and Hystem. In a diabetic mouse skin wound healing model, PGmatrix-encapsulated hADMSCs showed better healing at an early stage, suggesting possible improvement in anti-inflammatory and immunosuppressive effects. Hypertrophic epithelium re-epithelialization was observed among all three hydrogel-treated groups compared to the control group, but no significant difference in collagen density was found between the groups. 

## 2. Materials and Methods

### 2.1. Culture of hADMSCs

The 2D cell culture method was used for hADMSC population expansion. To ensure quality of hADMSCs (ATCC, Manassas, VA, USA) for subsequent experiments, all hADMSCs used were cultured for less than 5 passages. After thawing, hADMSCs were cultured in T150 flasks at 3000 cells/cm^2^ with MEM-alpha (low glucose, with Glutamax) (Thermo Fisher Scientific, Grand Island, NY, USA) containing 10% human Platelet Lysate (Stemcell Technologies, Vancouver, BC, Canada), heparin (Sigma-Aldrich, St. Louis, MO, USA), and 1% penicillin-streptomycin (Sigma-Aldrich); culture media were changed every other day. Cells were harvested when reaching 80% confluency. Culture media were removed from the flask and cells were then rinsed with Dulbecco’s phosphate-buffered saline (DPBS). To detach the hADMSCs from the culture flask, 0.05% trypsin-EDTA (Sigma-Aldrich) was added after removing DPBS and cells were incubated at 37 °C for 3–5 min followed by gentle tapping. Culture media, equal to the volume of trypsin solution, were then added to stop enzymatic reaction. The hADMSC suspension was transferred to 50 mL centrifuge tubes and centrifuged at 200× *g* for 5 min. The supernatant was discarded and cells were resuspended in fresh culture media or DPBS for downstream experiments.

### 2.2. Hydrogels and Cell Encapsulation

The cell density for each hydrogel encapsulation and the non-cross-linked HA gel was 6 × 10^6^ cells/mL. PGmatrix hydrogel (PepGel LLC., Manhattan, KS, USA) was prepared at 0.3% gel concentration (based on preliminary experiments; data not shown), following the manufacturer’s recommended method, by mixing the cell suspension in PGmatrix stock solution and PGwork, then incubating at 37 °C for gelation. For non-crosslinked HA gel [37], sodium hyaluronate powder (Sigma-Aldrich, St. Louis, MO, USA) was weighed and mixed with hADMSC suspension to reach a final 3% HA concentration, which is optimal according to Feng et al. [37]. For Hystem, the cell suspension was mixed with the Hystem hydrogel kit (Sigma Aldrich) components, Gelin-S, Glycosil, and Extralink, following the manufacturer’s instruction and incubated at 37 °C for 1 h for gelation. 

### 2.3. Cell Counting Kit-8 (CCK8)

Cell counting kit-8 (CCK8) assay is a colorimetric assay similar to MTT assay but has no cytotoxicity. The resulting optical density (OD) reflects cell metabolic activity level and cell viability. Wells in a 96-well plate were coated with 1% PGmatrix, followed by 30 min gelation at 37 °C, to prevent cell adhesion to plastic. hADMSCs encapsulated in 3% Hystem and 0.3% PGmatrix were plated into 3 wells for each treatment. Twenty microliters/well hADMSC-encapsulated 3% HA gel and hADMSC suspension were plated into 18 wells for each treatment. All plates were incubated at 37 °C for 1 h to allow complete gelation of PGmatrix and Hystem. CCK8 solution (Dojindo Molecular Technologies, Inc., Rockville, MD, USA) was mixed with DPBS at 1:9 ratio, 100 µL mixture/well (10 µL CCK8 solution for suspension wells) was added at 0, 3, 6, 12, 24, and 48 h after gelation and incubated for 1 h at 37 °C. Absorbance at 450 nm was measured with SpectraMax Plus Absorbance Microplate Reader (Molecular Devices, LLC., San Jose, CA, USA) right after CCK8 addition and after 1 h incubation. The final optical density (OD) of each well = OD after 1 h incubation—OD right after CCK8 addition. After measurement, the CCK8 mixture in Hystem and PGmatrix wells was replaced with DPBS. Wells containing 3% HA and suspension were discarded after measurement.

### 2.4. Cell Viability Assay

Acridine orange (AO) and propidium iodide (PI) dyes stain all cells and dead cells, respectively. Suspension of hADMSCs and hADMSCs encapsulated with 0.3% PGmatrix, 3% HA, and Hystem were prepared as described previously and loaded into 1 mL syringes. The syringes were incubated at 37 °C. To measure viability, a droplet of the sample was taken from each of the 4 groups. The droplets were added to 50 µL DPBS and pipetted to ensure even cell distribution. Forty microliters of the diluted samples were mixed with 40 µL AO/PI staining solution (Nexcelom Bioscience LLC., Lawrence, MA, USA). Then, 20 µL mixture (containing > 200 cells) was loaded into a counting chamber and counted with a Cellometer Auto2000 (Nexcelom Bioscience, Lawrence, MA, USA). Viability = live cell number/total cell number. Counting was performed in triplicates on each 80 µL cell-dye mixture. 

### 2.5. Immunostaining

Immunostaining was performed to assess expression level of pluripotency marker stage-specific embryonic antigen-4 (SSEA4) and organization of actin cytoskeleton. Hydrogel-encapsulated hADMSCs were prepared as described earlier and loaded into 1 mL syringes and 2D samples were prepared by plating hADMSCs suspended in media onto gelatin-coated glass coverslips. All samples were incubated at 37 °C overnight. Hydrogel encapsulated hADMSCs were transferred into 1.5 mL microtubes before overnight fixation of all samples using 10% neutral-buffered formalin. After fixation, samples were washed with wash buffer (DPBS without Ca^2+^/Mg^2+^ + Triton X-100 + gelatin from cold-water fish) (Sigma-Aldrich) and incubated with glutamine solution (Sigma-Aldrich) for 15 min to remove free aldehydes. The staining process was carried out using Invitrogen™ Pluripotent Stem Cell 4-Marker Immunocytochemistry Kit (Thermo Fisher) following manufacturer’s instruction. Primary antibodies were diluted with wash buffer, added to each sample, and incubated overnight at 4 °C. Samples were then rinsed with wash buffer to remove excess antibodies. Secondary antibodies were diluted, added to each sample, incubated for 3 h, and rinsed with wash buffer. Samples in microtubes were moved onto glass coverslips. 

To stain actin, Phalloidin-iFluor 594 (Abcam, Boston, MA, USA) and Deoxyribonuclease (DNase) I Alexa Fluor 488 Conjugate (Thermo Fisher) stock solutions were prepared according to manufacturers’ protocols. After removing free aldehydes, fixed samples were incubated with wash buffer for 5 min to increase permeability. Samples were then incubated with Phalloidin and DNase I working solution for 30 min, followed by rinsing with wash buffer to remove excess staining conjugates. One drop of 4′,6-diamidino-2-phenylindole (DAPI) solution (Abcam) was added to each sample and incubated for 10 min at room temperature. Glycerol was added to each sample before mounting the cell suspension. Images were acquired with a Zeiss 700 microscope (Carl Zeiss Microscopy, LLC., White Plains, NY, USA) and analyzed with Fiji-ImageJ [51].

### 2.6. Mass Spectrometry

Mass spectrometry (MS) was used to detect total proteins released by hADMSCs. To collect conditioned medium (CM) samples, an equal number of harvested hADMSCs were either plated into a T75 flask for 2D method or encapsulated in Hystem or 0.3% PGmatrix hydrogel then transferred into 50 mL centrifuge tubes. Cells in T75 flask were incubated at 37 °C overnight for attachment, while hydrogel-encapsulated samples in 50 mL tubes were incubated at 37 °C for 1 h to complete gelation. After incubation, hADMSC culture medium in T75 flask was removed and the cells were washed twice with DPBS and MEM-alpha medium was added to the flask. For hydrogel-encapsulated samples in 50 mL centrifuge tubes, the same amount of MEM-alpha medium was added into the two tubes. The samples with MEM-alpha medium were then incubated at 37 °C for 48 h before CM collection. Collected CM samples were filtered through 0.22 µm syringe filters (Corning) to remove cell debris.

The resulting filtrates were concentrated using 3 kDa Amicon Ultra-4 centrifugal filter units (MilliporeSigma, Burlington, MA, USA), washed with 1 × TBS (50 mM Tris-Cl, pH 7.6; 150 mM NaCl), and reconcentrated to remove phenol red. Total protein concentration of the concentrated samples was determined with a BCA protein assay kit (Thermo Fisher). Protein digestion, peptide separation, and MS were performed as follows: 50 μg total proteins were digested with sequencing grade modified trypsin (Promega Corporation, Madison, WI, USA) at a ratio of 10:1 on S-Trap^TM^ column (ProtiFi, Farmingdale, NY, USA). After incubation overnight at 37 °C, proteins were eluted from the column sequentially with 50 mM ammonium bicarbonate, 0.2% formic acid in water, and 50% acetonitrile in water. Eluents were pooled, dried with speed-vac, and resuspended in 50% acetonitrile in water supplemented with 0.1% formic acid. Peptide separation was performed with ACQUITY M-Class UPLC (Waters Corporation, Milford, MA, USA) on an HSS T3 Column (Waters, 100 Å, 1.8 μm, 300 μm × 150 mm). Elution was performed at 2 μL/min with a gradient of acetonitrile (containing 0.1% formic acid) from 5% to 50% in water (containing 0.1% formic acid) over 120 min, followed by an isocratic gradient of 50% acetonitrile in water for an additional 10 min. Detection was performed with Xevo G2-XS Q-Tof mass spectrometer (Waters) coupled with NanoLockSpray ionization source, operated under MS^E^ mode for data-independent acquisition. 

Raw data were processed using Progenesis QI for Proteomics (Waters) by manually specifying the threshold intensities at 250 and 150 counts for low and elevated energies, respectively. Peptides containing more than 6 residues were selected and identified using UniProt proteome (ID: UP000005640) as the search database. Criteria for protein identification search were set as follows: two possible missed cleavages using trypsin and two modifications including one fixed cysteine carbamidomethylation and one variable methionine oxidation. Ion matching requirements were set at fragments/peptides ≥ 3, fragments/protein ≥ 7, and peptides/protein ≥ 1. Fold changes of protein expression levels from different CM samples were calculated using the Experiment Design Function in Progenesis QI for Proteomics. 

### 2.7. Enzyme-Linked Immunosorbent Assay (ELISA) for Human Vascular Endothelial Growth Factor A (VEGF-A)

Nine transwells in a 24-well plate (Corning Inc., Corning, NY, USA) were coated with 50 µL 0.5% PGmatrix or Hystem hydrogels and incubated at 37 °C for 1 h for gelation. Three wells of approximately 1.5 × 10^4^ hADMSCs/well were encapsulated in 3% HA, Hystem, and 0.3% PGmatrix following methods described previously. The samples were then incubated at 37 °C for 1 h to allow gelation. Another 3 wells of suspended hADMSCs were plated directly onto transwells and 1 mL of MEM-alpha medium was added to each well. One hundred fifty microliters of CM was collected from each well at 6, 24, 48, and 72 h after gelation. The CM samples were centrifuged at 350× *g* for 10 min to remove cell debris, then stored at -20 °C until the ELISA was performed. To detect VEGF-A, a precoated VEGF-A ELISA kit (Thermo Fisher) was used following manufacturer’s instructions. Briefly, the plate was washed with wash buffer and dried before adding standard diluents and samples in duplicate. The plate was then covered and placed on a microplate shaker (400 rpm) at room temperature for 2 h. After shaking, the plate was emptied and washed 6 times with wash buffers. Then 100 µL Biotin Conjugate was added to each well, the plate was covered and placed on the shaker at room temperature for 1 h, followed by another set of 6 washes. Following this, 100 µL diluted Streptavidin-HRP was added to all wells, including blanks, and the plate was placed on the shaker for 1 h, at room temperature. The plate was emptied and washed 6 times before adding 100 µL TMB Substrate Solution. The plate was then incubated at room temperature in the dark for 30 min and 100 µL stop solution was quickly added to all wells at the end of the incubation; absorbance was measured at 450 nm with SpectraMax Plus Absorbance Microplate Reader (Molecular Devices, LLC.).

### 2.8. Diabetic Mice Wound Healing Model

All animal procedures were conducted in accordance with institutional guidelines using a protocol approved by the Institutional Animal Care and Use Committee (IACUC) at Kansas State University (Protocol #4332, Date of approval 21 August 2020). Thirty-two male CD-1 mice aged 7–8 weeks were obtained from Charles River Laboratory and housed individually at Kansas State University. Mice were housed individually with free access to food and water and under 12 h light/12 h dark cycles for one week before diabetes induction. Streptozotocin (STZ) (Sigma-Aldrich) was dissolved in physiological saline at 4 mg/mL (Baxter, Deerfield, IL, USA), sterile filtered with 0.22 µm syringe filter (Corning) into sterile glass vials and kept in dark on ice. STZ solution was made right before each injection. The mice received 5 daily intraperitoneal injections of STZ solution at 40 mg/kg body weight, with 10% sucrose (Sigma-Aldrich) added in drinking water for 5 days to induce diabetes. Blood glucose was monitored with Contour Next One whole blood glucose meter (Ascensia Diabetes Care, Parsippany, NJ, USA); a mouse with a blood glucose > 250 mg/dL was considered diabetic. Ten days after the last STZ injection, mice were anesthetized with isoflurane inhalation. An 8 mm round full-thickness excisional wound was created by sharp dissection on the dorsolateral thoracolumbar region of each mouse. A sterilized stainless-steel wound splint (Menards, Eau Claire, WI, USA) was placed on each wound and secured with 4-0 nylon sutures (MWI, Boise, ID, USA) to the skin edges to minimize wound contraction. An injection of either DPBS control or one of the encapsulated hydrogels was applied directly into the skin edges immediately after splint suturing was completed. Mice were randomly assigned to one of the following treatment groups: 100 µL DPBS (Sigma-Aldrich); 6 × 10^5^ hADMSCs encapsulated in 100 µL 0.3% PGmatrix; 6 × 10^5^ hADMSCs encapsulated in 100 µL Hystem hydrogel; 6 × 10^5^ hADMSCs encapsulated in 100 µL 3% HA. Each encapsulation was incubated at 37 °C overnight. Mice were monitored for 21 days after the surgery with photos taken immediately following surgery (day 0) and on days 4, 7, 10, 14, and 21. Photos of the wound were analyzed with Fiji-ImageJ. Wound outlines were carefully traced and the surface area within the outlines was measured to track wound area change during healing. 

### 2.9. Histological Analysis

All mice were euthanized in a CO_2_ chamber 21 days post-surgery and tissue from the wound area was excised with scissors and fixed in 10% neutral-buffered formalin (Sigma-Aldrich). Tissue samples were processed in paraffin, sectioned at 4 µm, and stained with hematoxylin and eosin (H&E) and Masson’s trichrome, following standard procedures, by the histology laboratory at the Kansas State Veterinary Diagnostic Laboratory. Healing was assessed by a boarded anatomic pathologist, in a single blind study design, and a subjective scoring system was used to assess the skin sections. Score 1 represents re-epithelialization with relatively “thin” epithelium (< 6 cell layers thick); Score 2 represents re-epithelialization with hypertrophic epithelium; Score 3 represents thick re-epithelialization with basement membrane (dermal-epidermal junction) clefting and/or ulceration. Collagen deposition was evaluated in Masson’s trichrome-stained sections using Fiji-ImageJ, following the method described by Chen et al. [52]. Collagen intensity was converted to optical density (OD) using formula OD = log_10_(Max Intensity/Mean Intensity), in which Max Intensity = 255, Mean Intensity = Total integrated collagen density/total area measured. The formula was derived from a method described by Wieck et al. [53].

### 2.10. Statistical Analysis

Statistical analysis was performed with JASP [54]. Data measured at multiple timepoints, including CCK8, viability, ELISA, and wound healing, were analyzed with repeated measure ANOVA with α level ≤ 0.05 as statistical significance. Fluorescence intensity of SSEA4 was compared using Student’s t-test (two-tailed) with α level ≤ 0.05 as statistical significance. All data are shown as the means ± standard error of the mean (SEM). * *p* < 0.05, ** *p* < 0.01, *** *p* < 0.001.

## 3. Results

### 3.1. PGmatrix Significantly Extended hADMSC Survival after Encapsulation 

The effect of using different hydrogels on hADMSC viability and metabolism over 48 h after encapsulation was measured with a CCK8 assay (Figure 1A). Cells had very limited nutrient supply since the only medium added was that used during encapsulation. Results showed that the optical density (OD) values of hADMSCs encapsulated in 0.3% PGmatrix and Hystem remained high in the first 12 h and then slowly decreased up to 48 h, but the OD with Hystem was much lower right after encapsulation (0 h) than with the 0.3% PGmatrix and suspension groups (Figure 1A). The OD values of hADMSCs encapsulated in 3% hyaluronic acid (HA) remained significantly lower than the other two hydrogel groups and the control group during the first 12 h (*p* < 0.05) and reached a similar level as other groups by 48 h. Suspension of hADMSCs showed high OD readings comparable to PGmatrix and Hystem groups during the first 12 h but dropped to a level significantly lower than the PGmatrix group (*p* = 0.003) 24 h after encapsulation (Figure 1A).

According to the principle of the CCK8 assay, the OD readings obtained reflected combined results from cell viability and metabolism. To better estimate the cell viability change after encapsulation, we took a droplet from each test sample at each time point and counted live/dead cells with a cellometer (Figure 1B). During the first 6 h after encapsulation, viability of the Hystem group decreased the most rapidly, significantly lower than other groups (*p* < 0.001). After 12 h, only PGmatrix group maintained high viability above 80%, significantly higher than the suspension group (*p* = 0.048) and the Hystem and 3% HA groups (*p* < 0.001). All groups dropped to 50–60% viability after 24 h. The 0.3% PGmatrix group maintained viability at 46.3% at 48 h after encapsulation, while the other two encapsulation groups decreased to below 40%. The suspension group had the lowest viability at 7.3%, followed by Hystem group at 18.6%. The 0.3% PGmatrix hydrogel demonstrated superior protective effects on hADMSCs. Although the Hystem group had a lower percent viability, cell metabolism was enhanced, with the Hystem group having OD readings comparable to the PGmatrix group during the first 12 h.

### 3.2. PGmatrix Was Superior to Maintain SSEA4 Surface Antigen in hADMSCs

A subpopulation of MSCs express pluripotency markers, such as SSEA4; these SSEA+ cells were shown to have higher differentiation potential [55]. To examine SSEA4 production, immunostaining was performed on hADMSCs encapsulated in different hydrogels after overnight incubation at 37 °C overnight to detect for possible upregulation of stem cell markers; 2D-plated hADMSCs were used as control. The 0.3% hADMSCs encapsulated in PGmatrix showed a high level of SSEA4 fluorescent immunolabeling (Figure 2A). The quantified SSEA4 intensity of 0.3% PGmatrix group was significantly higher than all other groups (Figure 2B). 

In addition, immunolabeling for cytoskeleton filaments, F-actin and G-actin, was performed on these hADMSCs. We found that though the release of actin tension, reflected by a reduction in F-actin stress fiber, existed in all 3D-encapsulated hADMSCs, actin organization is slightly different in hADMSCs in different hydrogels (Figure 3A). In 3% HA, F-actin formed various small clusters dispersing within and on the edge of hADMSCs. Similar F-actin distribution was observed in hADMSCs treated with 0.3% PGmatrix as 3% HA, except F-actin clusters were larger in PGmatrix-encapsulated hADMSCs. In contrast, F-actin in Hystem-encapsulated hADMSCs formed bright cortical filaments to support cell shape, with the rest relatively evenly distributed inside the cells. G-actin in these groups followed similar patterns as F-actin, with PGmatrix group having the highest level of actin agglomeration. Different from Zhou et al. [19], we observed bright F-actin clusters when F-actin stress fiber reduced in 3D. Assessment of F-actin fluorescence intensity showed no difference between the groups (Figure 3B), indicates changes in F-actin organization, not F-actin protein production.

### 3.3. hADMSCs in PGmatrix Secreted High-Immune-Responsive Proteins and Comparable Levels of VEGF-A and TGFβ1 Factors Compared to Other Hydrogels

To obtain comprehensive understanding of the hADMSC secretome in 2D and 3D PGmatrix and HA-based hydrogel Hystem, mass spectrometry (MS) was performed on conditioned medium (CM), collected after incubating encapsulated hADMSCs for 48 h. Out of the 1013 proteins identified with MS, 99 proteins were detected with high confidence (≥ 10 unique peptides identified). The 0.3% PGmatrix and Hystem samples had the highest abundance in 32 and 23 of the 99 proteins, respectively. The remaining 44 proteins were the most abundant in the 2D sample. Interestingly, we found that most extracellular matrix (ECM) and adhesion-related proteins, including collagen alpha-1(VI) chain (COL6A1), fibronectin (FN1), Laminin beta-1 (LAMβ1), secreted protein acidic and rich in cysteine (SPARC), and insulin-like growth factor-binding protein 7 (IGFBP7), were high in 2D and Hystem samples and low in the 0.3% PGmatrix sample (Figure 4A), while immune-response-related proteins, including semaphoring-7A (SEMA7A), alpha-1-antitrypsin (SERPINA1), apolipoprotein A-I (APOA1), and galectin-3-binding protein (LGALS3BP) were often the highest in the 0.3% PGmatrix sample and least abundant in 2D samples (Figure 4B). Intracellular proteins, including elongation factor 1-gamma (EEF1G), heat shock protein 90-beta (HSP90AB1), plastin-3 (PLS3), and annexin (ANXA2) were generally more abundant in Hystem samples (Figure 4C), suggesting loss of cell membrane integrity and cell death in Hystem samples.

In addition to cell properties, factors secreted by hADMSCs have been shown to play important roles in their therapeutic potentials [56,57]. The MS data also showed vascular endothelial growth factor receptor 1 (VEGFR-1/FLT1) and transforming growth factor beta-1 (TGFβ1). Hystem-encapsulated hADMSCs released more VEGFR-1, while PGmatrix-encapsulated hADMSCs secreted slightly more TGFβ1 (Figure 4D). As VEGFR-1 is involved in regulating vascular endothelial growth factor (VEGF) level, its production and secretion depends on the extent of VEGF signaling needed [58]. To confirm if VEGFR-1 would reflect a similar level of VEGF, an ELISA assay was used to determine VEGF-A released by hADMSC under 2D matrix or 3D encapsulation with different hydrogels at different culture time points up to 72 h (Figure 4E). Hystem and 0.3% PGmatrix groups had no significant differences in VEGF-A release; in contrast, the 2D group released significantly more VEGF-A after 48 h incubation. However, 3% HA showed a slight increase in VEGF-A release level after 72 h incubation. 

### 3.4. PGmatrix Encapsulation Promoted Healing at Early Stage

A diabetic mice wound skin model was used to assess the effect of different hydrogels on the therapeutic effect of hADMSC 3D encapsulations. Diabetes was induced in CD-1 male mice by five daily injections of streptozotocin (STZ). Ten days after the last STZ injection, 8 mm round full-thickness excisional wound was created on the dorsum of each mouse with a sterilized steel wound splint attached to minimize contraction (Figure 5A). After splint attachment, mice were randomly assigned to receive injection of DPBS as control, hADMSCs encapsulated in 3% HA, and 0.3% PGmatrix or Hystem hydrogels incubated at 37 °C overnight. The healing process was tracked for 21 days. The 0.3% PGmatrix group had significantly better healing than the HA and control group in terms of wound area by day 4 after the surgery (Figure 5A,B). Then, no significant difference was seen among the three hydrogels by day 7 and 10, though all of them showed better healing than the control. However, all groups reached similar healing levels after 14 days, as the effect of healing contracture was observed to became prominent after day 14 and no difference was identified between groups by 21 days (Figure 5A,B).

All mice were euthanized for histologic analysis on day 21. The excised wound samples were fixed and stained with hematoxylin and eosin (H&E) and Masson’s Trichrome (Trich) (Figure 6A). The images were assessed by a boarded anatomic pathologist, in a single blind study design, and scored for re-epithelialization from 1 to 3, with 1 being the closest to healed epithelium. Of all samples evaluated, all hydrogel groups had a similar level of re-epithelialization (Figure 6B). Although some studies have shown that MSCs would promote collagen expression and deposition during wound healing [37,59,60], a significant difference among the hydrogel groups was not detected in this study (Figure 6C).

## 4. Discussion

Mesenchymal stem cells and their secretome have been gathering attention for various therapeutic effects [61,62]. Some studies intended to use the MSC secretome only to avoid risks related to cell transplant [23,34], but the use of cells may be more beneficial than the secretome alone, as MSCs might be able to reside and differentiate into target tissue [62]. In clinical settings, it is difficult to ensure transplant with freshly harvested cells. A benefit of MSCs it that they can be prepared either with frozen stock or with biomaterials as a delivery carrier.

Hyaluronic acid (HA) and HA-based hydrogels, such as Hystem have been used extensively in MSC delivery studies and have shown varied levels of beneficial effects [34,37,63,64]. However, these hydrogels are still relatively hard to handle, which hinders their application in clinical settings. We developed a shear-thinning peptide hydrogel (PGmatrix) [48,65] and used it successfully for virus delivery [49] and 3D culture of hiPSCs [47]. The main peptide in PGmatrix was made of 19 amino acids rationally designed by utilizing human muscle and spider silk protein domains [47] and it provided mainly mechanical support for cells without significant chemical or biological cues [47]. PGmatrix was also identified to have minimal cell damage from mechanical stress through pipetting or extrusion printing nozzle [47] and high in vivo injection biocompatibility [50]. Thus, PGmatrix may be an alternative to HA or HA-derived hydrogels for in vivo delivery of MSCs.

We tried to mimic a real-world setting in this study. Syringes loaded with encapsulated hADMSCs were usually prepared hours before treatment with no access to fresh culture media during this process. Based on all the data collected in this study, PGmatrix significantly extended hADMSC survival after encapsulation, while HA-based Hystem seemed to stimulate cell metabolic activity. These results are probably linked to different ways the hydrogels interact with hADMSCs after encapsulation. SSEA4 is commonly used as a marker protein for undifferentiated human pluripotent stem cells but is also synthesized by a subpopulation of MSCs from several tissue sources [66]. SSEA4^+^ hADMSCs were reported to have better endothelial and osteogenic differentiation potential [55]. Although we observed elevated expression of SSEA4 in hiPSCs cultured in PGmatrix [47], which was difficult to interpret due to the high inherent SSEA4 level in hiPSC, the results from this study showed that PGmatrix upregulated SSEA4 in hADMSCs, as compared to HA hydrogels (Figure 2). Since PGmatrix does not contain cell adhesion sites, the SSEA4 upregulation effect might relate to mechanical cues provided by the PGmatrix environment. According to actin staining images (Figure 3A) and intensity quantification (Figure 3B), hADMSCs had very bright actin agglomeration and much dimmer actin in other parts of the cells when encapsulated in PGmatrix than in HA or Hystem. Researchers have found that when in 3D configuration, MSCs can cluster together to form spheroids [33,37,67]. According to Zhou et al. [19], MSCs in 2D culture exhibited a high amount of F-actin stress bundles across the cell body. After being moved into 3D configuration, most intracellular stress fiber was replaced with thin filaments and spots of actin clusters. Expression level of β-actin reduced during this transition, releasing actin cytoskeleton tension. Relaxation of actin tension would improve MSC stemness and therapeutic effects [19,37,68,69]. Though we did not detect a change in F-actin intensity between 2D and 3D, formation of more compact actin clusters in PGmatrix-encapsulated hADMSCs may explain the SSEA4 upregulation effect and suggested better stemness of these hADMSCs compared to those in suspension or encapsulated in HA hydrogels.

In addition to improved viability and stemness, cell secretome is also considered the key to hADMSC therapeutic effects. According to MS analysis, hADMSCs encapsulated in PGmatrix had higher release levels than Hystem-encapsulated hADMSCs for most of the immune-responsive proteins detected, not limited to SEMA7A [70,71], SERPINA1 [72], APOA1 [73], LGALS3BP [74] (Figure 4C), as well as Alpha-2-macroglobulin [75], plastin-2 [76], and immunoglobulins (IGHM, IGHG1, IGKC) (Supplement Appendix A). This showed that PGmatrix had a stronger stimulation effect on the release of immune-responsive proteins. In addition, hADMSCs in all three hydrogels had comparable VEGF-A release according to ELISA assay and similar TGFβ1 release was found in the medium according to MS analysis (Figure 4D,E). Similar release of VEGFR-1 was observed from MS analysis. Because VEGFR-1 is involved in regulating VEGF levels, its production and secretion depend on the extent of VEGF signaling needed [58]. TGFβ1 was reported to have pro-inflammatory effects [77,78], but 3D-encapsulated MSCs were found to have better anti-inflammation effects [33,35,37], which are in good agreement with lower TGFβ1 release in Hystem and PGmatrix-encapsulated hADMSCs than in the 2D matrix.

MS analysis also revealed that hADMSCs in Hystem, as compared to PGmatrix, showed higher abundance of most ECM proteins, including collagen, fibronectin, and laminin, as well as proteins related to cell adhesion and motility, including SPARC [79] and IGFBP7 [80] (Figure 4A). Furthermore, abundant intracellular proteins that are not expected to be secreted into the CM, such as EEF1G [81], PLS3 [82], and HSP90AB1 [83], were found in the CM of the Hystem hADMSCs (Figure 4C). Together with the high level of membrane protein annexin (ANXA2) detected in the CM of Hystem hADMSCs, this suggests that a large number of cells from hADMSCs in Hystem might have experienced loss of cell membrane integrity or cell death that caused release of intracellular and membrane materials. This is consistent with the results from CCK8 and viability assay, in which cell viability could not be well maintained after encapsulation with Hystem with limited medium. This would limit the practical use of Hystem when high cell viability is paramount.

With in vitro analysis, PGmatrix is superior to HA or HA-based hydrogels in terms of maintaining cell viability in limited culture medium and releasing more immune-responsive proteins. In our in vivo study, PGmatrix improved wound healing in the early stage starting at day 4 and then became similar to HA and Hystem by day 7 up to day 10 (Figure 5). All hydrogel groups accelerated wound healing before day 10 compared to the control; however, no significant difference was observed among all groups by day 14 and all achieved complete healing, through gross examination, by day 21 (Figure 5). According to the literature [37,59,60], re-epithelialization and collagen deposition would favor wound healing. However, no significant differences in re-epithelialization or collagen density were observed among the treated or control groups (Figure 6B,C). The healthy outbred mice were chosen with the purpose of better representing a heterogeneous population compared to inbred diabetic strains; however, the diabetes situation induced with STZ in the healthy outbred mice did not provide significant effects of slower healing complexity as expected by day 21, although all mice exhibited diabetic symptoms before wound creation. Nevertheless, healing differences were obviously observed at earlier timepoints, such as day 4 (Figure 5B), and our MS analysis data (Figure 4B) suggested that PGmatrix encapsulation promoted release of immune-responsive proteins, which would help reduce inflammation and contribute to faster wound healing, as observed in the early stage of the healing process. For more complex and larger wounds, the promotional healing effects of encapsulated hADMSCs might be more observable, as reported in some chronic, nonhealing wounds [59]. In summary, pretreatment of MSCs with hydrogel encapsulation may be better suited for use in promoting anti-inflammatory and immunosuppressed environments and, thus, suitable for use in inflammatory and autoimmune diseases. PGmatrix encapsulation would be a better alternative to HA hydrogels as it improves maintenance of cell viability. Nonetheless, choosing optimal MSC delivery hydrogels merits further exploration.

## 5. Conclusions

PGmatrix-encapsulated hADMSCs showed superior cell viability maintenance, higher SSEA4 expression, and increased release of factors related to the immune response. All three hydrogel-encapsulated hADMSCs had similar secretion levels of VEGF-A and TGFβ1. Beneficial effects of PGmatrix-encapsulation on diabetic skin wounds were observed during the early stage of healing. Together, our research suggested that encapsulation with PGmatrix or HA-based hydrogels as pretreatment and delivery method for MSCs may be suitable for inflammatory or autoimmune disease models, while PGmatrix encapsulation would be better than HA-based hydrogels for tissue regeneration, as MSC viability may be more important. Peptide hydrogel PGmatrix performed equally well or better than HA-based Hystem in all tests, rising as a new candidate material for therapeutic cell delivery.

## Figures and Tables

**Figure 1 biomolecules-12-01317-f001:**
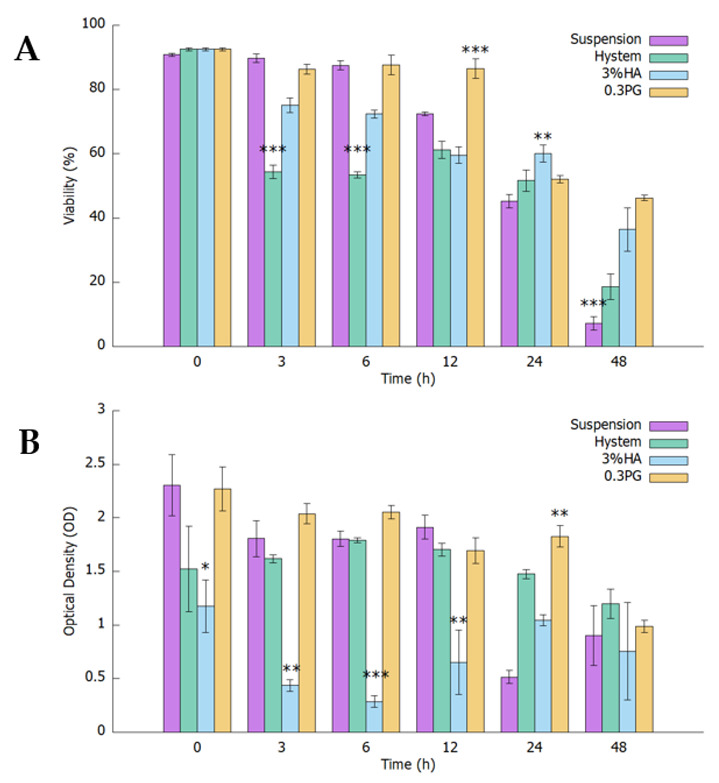
After encapsulation with no culture medium provided, CCK8 and viability of hADMSCs in different 3D conditions measured at different time points. (**A**) CCK8 results were recorded as increase in optical density (OD) at each time point (OD after 1 h CCK8 incubation—OD right after CCK8 addition); (**B**) Percent viability at each time point measured with cellometer. All results are shown as mean ± standard error of the mean (SEM), * *p* < 0.05, ** *p* < 0.01, *** *p* < 0.001.

**Figure 2 biomolecules-12-01317-f002:**
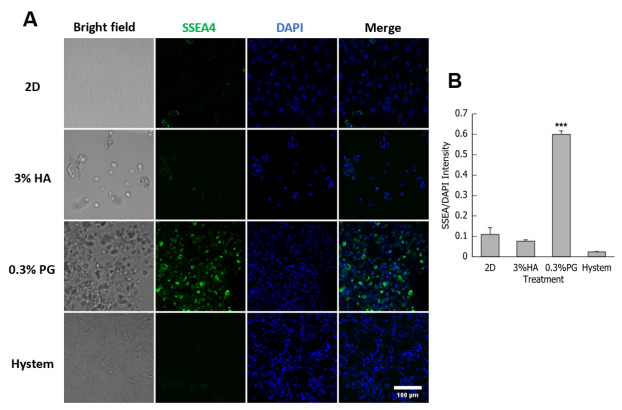
Encapsulated and 2D-plated hADMSCs with stage-specific embryonic antigen-4 (SSEA4) immunostaining and nuclear staining (DAPI). (**A**) Confocal images of cells, scale bar = 100µm; (**B**) Fluorescence intensity, shown as SSEA4/DAPI. Results are shown as mean ± SEM, *** *p* < 0.001.

**Figure 3 biomolecules-12-01317-f003:**
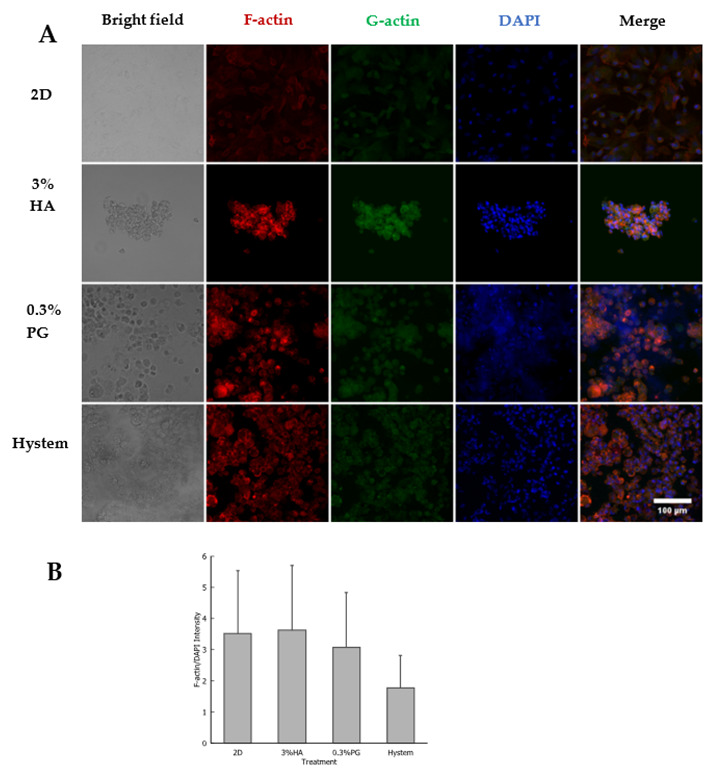
(**A**) Immunostaining of F- and G-actin in hADMSCs encapsulated in different materials and incubated at 37 °C overnight. Nucleus was labeled with DAPI. Scale bar = 100µm. (**B**) F-actin signal intensity shown as F-actin/DAPI. Results are shown as mean ± SEM.

**Figure 4 biomolecules-12-01317-f004:**
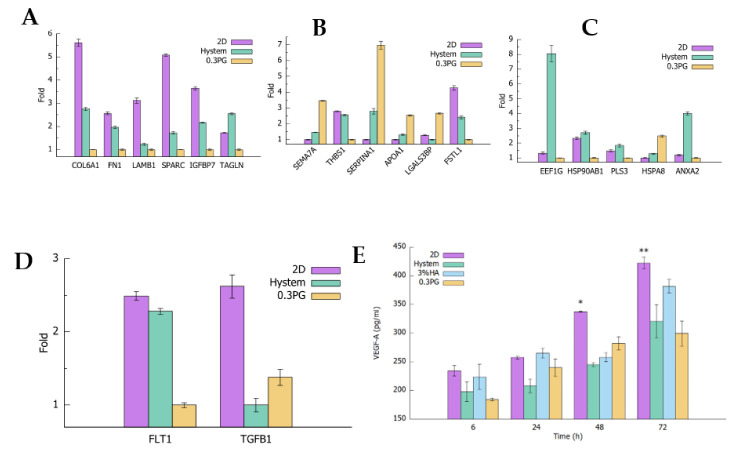
Analysis of the secretome of hADMSCs in Hystem and 0.3%PG encapsulations and 2D matrix. (**A**–**D**) Mass Spectrometry results of protein abundance expressed as fold change using the sample with lowest abundance of each protein as the reference. Representative proteins identified with high confidence (≥ 10 unique peptides) are grouped into 3 categories: extracellular matrix (ECM) and ECM binding proteins (**A**); immune-response-related proteins (**B**); intracellular proteins (**C**) Folds of Vascular endothelial growth factor receptor-1 (VEGFR1/FLT1) and transforming growth factor beta-1 (TGFβ1) are shown in (**D**,**E**) Levels of VEGF-A released by hADMSCs in 2D or 3D encapsulation measured with ELISA. Results are shown as mean ± SEM, * *p* < 0.05, ** *p* < 0.01.

**Figure 5 biomolecules-12-01317-f005:**
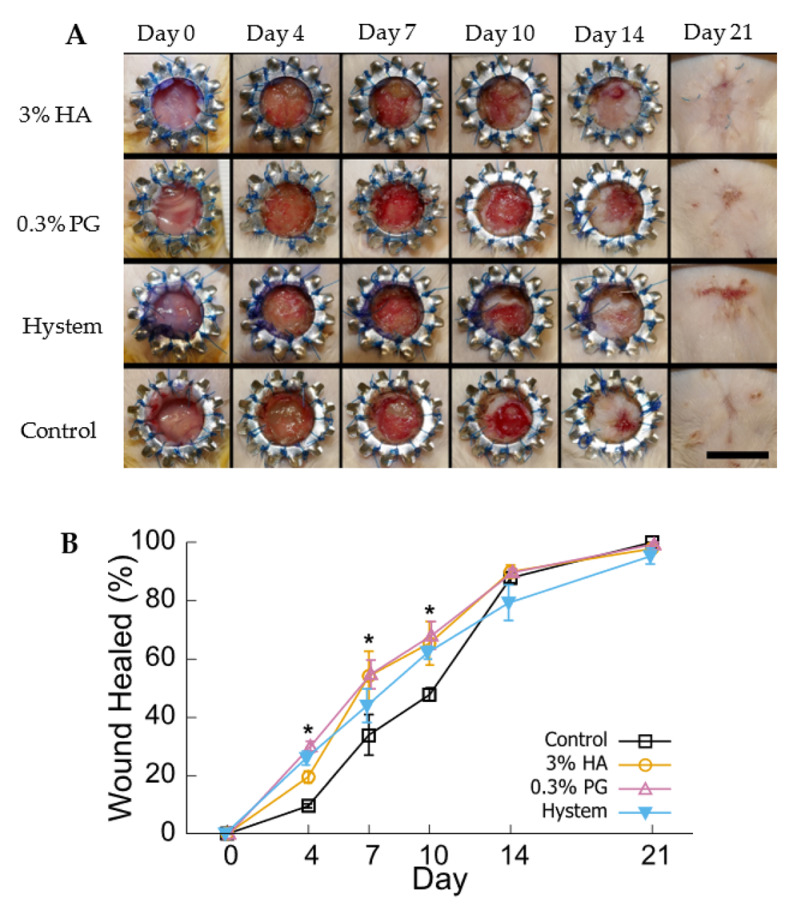
Skin wound healing in diabetic mice. (**A**) Photos of the wound area taken after surgery (day 0) and 4, 7, 10, 14, and 21 days after surgery. Scale bar = 10 mm. (**B**) Quantification of wound surface healing. Wound healed (%) = 100—wound area/ Day 0 wound area * 100. Results are shown as mean ± SEM, * *p* < 0.05.

**Figure 6 biomolecules-12-01317-f006:**
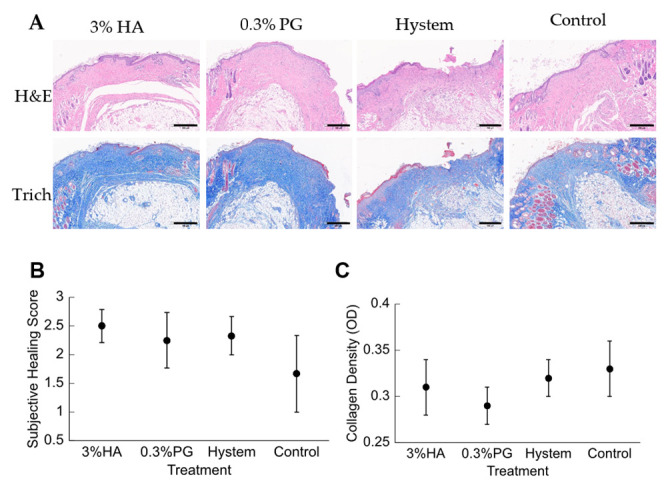
Histologic analysis of mouse cutaneous wound samples. (**A**) Hematoxylin and eosin (H&E) and Masson’s Trichrome (Trich)-stained samples from each treatment group. Scale bars were shown on the bottom right of every image. Scale bar for 3% HA and Control = 500 µm, scale bar for 0.3% PG = 600µm, scale bar for Hystem = 700 µm. (**B**) Re-epithelialization scores for each treatment group. Results are shown as mean ± SEM. (**C**) Collagen density from each treatment group expressed in optic density (OD). Results are shown as mean ± SEM.

## Data Availability

The data presented in this study are available on request from the corresponding author.

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
