# Peer review of "Injectable Peptide Hydrogel Encapsulation of Mesenchymal Stem Cells Improved Viability, Stemness, Anti-Inflammatory Effects, and Early Stage Wound Healing"

_biomolecules, 2022, doi:10.3390/biom12091317_

Round 1
Reviewer 1 Report (New Reviewer)
Title - Injectable Peptide Hydrogel Encapsulation of Mesenchymal 2 Stem Cell Improved Viability, Stemness, and Anti-Inflamma-3 tory Effects
Journal – Biomolecules
Manuscript Number: biomolecules-1854688
Detailed Review Comments
Shear thinning Injectable Peptide Hydrogel was used by Li et al. to encapsulate mesenchymal stem cells. In the peptide hydrogels, the author says that MSCs have improved viability, stemness, and anti-inflammatory effects; nevertheless, I have some observations that I would like to share below.
1. Figure 1B. Why is the percentage of cell viability on day 0 lower than 10 percent? In the section under "Methods," you, the author, need to include the formula for calculating the percent viability.
2. Figure 1A Plotted optical density, and figure 1B shows the percent viability. It is not quite obvious if any of these figures is displaying unique information or the same information. The OD value shown in Figure 1A may have been used by the author in the calculation of the percentage of viability shown in Figure 1B.
3. In the presentation of the data shown in Figure 1, the authors have not addressed the question of why one material behaves better than the others.
4. The authors might want to think about presenting figure 1 as a bar graph instead of a line graph. I have faith that a greater amount of data may be comprehended from its bar graph depiction. In addition, statistical evidence (asterisk) can be supported.
5. Doesn't the fact that the level of cell viability is decreasing to below 50 percent in all of the groups imply that the material being utilized has a harmful effect? How can the author say that the viability has been increased in the title? Could you elaborate?
6. Figures 2 and 3. Illustrate the reasons why the confluency of cells in 3 percent HA is lower than in 0.3 percent PG (see the bright field). Figure 3 must also have the same labeling if that can be done.
7. If a concentration of HA that is 3 percent is hazardous to the cell, then why did the authors choose this concentration for their experiment? Could you elaborate?
8. Why did the authors get rid of the 3% HA group in Figure 4? The author should include the data for HA as well for the comparison.
9. The assertion made in the section title for 3.3 does not accord with the findings presented in Figure 4.
10. Make sure the labels are correct in Figure 5A.
11. The author is responsible for providing evidence of the materials' in vivo biocompatibility. In light of the fact that the in vitro results showed high toxicities, a subcutaneously implanted material biocompatibility study can be carried out.
12. If, on day 21, all of the animals had the same level of recovery, then what difference does it make if the PG Matrix material initially boosted healing?
13. The information in Figure 6 suggests that all of the materials are of equal quality. For this reason, I feel that the title of the article would lead the readers astray. Take into consideration altering the title of the manuscript.
14. Author might want to consider reworking the part about the conclusion.
I am certain that if authors include the aforementioned changes, not only will it improve the quality of the work, but it will also be beneficial to the reader. In conclusion, the work, in its current state, is not appropriate for publishing in Biomolecule and requires a significant amount of editing as well as rewriting.
Author Response
Shear thinning Injectable Peptide Hydrogel was used by Li et al. to encapsulate mesenchymal stem cells. In the peptide hydrogels, the author says that MSCs have improved viability, stemness, and anti-inflammatory effects; nevertheless, I have some observations that I would like to share below.
1. Figure 1B. Why is the percentage of cell viability on day 0 lower than 10 percent? In the section under "Methods," you, the author, need to include the formula for calculating the percent viability.
Answer: Thank you for your comments. The lowest viability obtained during the first 24 hours was about 50% from suspension sample, and no sample has less than 10% viability on day 0. We have added the formula for viability calculation (blue text) on page 3, line 146: “Viability = live cell number / total cell number”
2. Figure 1A Plotted optical density, and figure 1B shows the percent viability. It is not quite obvious if any of these figures is displaying unique information or the same information. The OD value shown in Figure 1A may have been used by the author in the calculation of the percentage of viability shown in Figure 1B.
Answer: Thank you for the comments. We have described the methods for OD (CCK8 assay) and viability (AO/PI staining, counting with cellometer) in two different sections in the Method section on Page 3 Line 123 and Page 3 Line 138, respectively. Viability was not calculated from OD. For viability calculation, please refer to our answer to question 1.
3. In the presentation of the data shown in Figure 1, the authors have not addressed the question of why one material behaves better than the others.
Answer: Thank you for the valuable opinion. As mentioned in Method – CCK8 assay (page 3) and Results (page 6), according to the mechanism of this CCK8 assay, the OD values are considered the combined results of cell metabolism and viability. Since different hydrogel would interact with hADMSCs differently, which could result in different metabolic effects. That is why we measured viability in a separate independent experiment in order to reveal the cell metabolic activity levels, allowing us to observe the significant effects of different materials on cells. This project is more application oriented, thus we did not explore in detail on the interaction between hADMSC and different hydrogels. It would be meaningful to further investigate the mechanism in the future.
4. The authors might want to think about presenting figure 1 as a bar graph instead of a line graph. I have faith that a greater amount of data may be comprehended from its bar graph depiction. In addition, statistical evidence (asterisk) can be supported.
Answer: It is a good suggestion. We have changed the figures into bar graphs and revised figure 1 caption to distinguish between CCK8 and viability measurements.
5. Doesn't the fact that the level of cell viability is decreasing to below 50 percent in all of the groups imply that the material being utilized has a harmful effect? How can the author say that the viability has been increased in the title? Could you elaborate?
Answer: Thank you for the comments. In discussion (page 15, line 532), we explained that “We tried to mimic real world setting in this study. Syringes loaded with encapsulated hADMSCs were usually prepared hours before treatment with no access to fresh culture media during this process.” The hADMSCs in this study were encapsulated at high density (6x106 cell/ml) with
very limited media supply, this would greatly impact cell survival. On the other hand, all hydrogels used in this project have good cytocompatibity for cell cultures or encapsulation, and were investigated in previous studies and showed no cytotoxicity at all. Below are some of the references cited in this article showing their biocampatibility:
3% HA
37. Feng, J.; Mineda, K.; Wu, S.H.; Mashiko, T.; Doi, K.; Kuno, S.; Kinoshita, K.;
Kanayama, K.; Asahi, R.; Sunaga, A.; et al. An injectable non-cross-linked hyaluronic-acid gel containing therapeutic spheroids of human adipose-derived stem cells. Sci. Rep. 2017, 7, 1548, doi:10.1038/s41598-017-01528-3.
Hystem
38. Zarembinski, T.I.; Skardal, A. HyStem®: A Unique Clinical Grade Hydrogel for Present and Future Medical Applications. Hydrogels - Smart Mater. Biomed. Appl. 2018, doi:10.5772/INTECHOPEN.81344.
PGmatrix
50. Carter, T.; Qi, G.; Wang, W.; Nguyen, A.; Cheng, N.; Ju, Y.M.; Lee, S.J.; Yoo, J.J.;
Atala, A.; Sun, X.S. Self-Assembling Peptide Solution Accelerates Hemostasis. Adv. Wound Care 2021, 10, 191–203, doi:10.1089/WOUND.2019.1109/ASSET/IMAGES/LARGE/WOUND.2019.1109_FIGURE9.JP EG.
6. Figures 2 and 3. Illustrate the reasons why the confluency of cells in 3 percent HA is lower than in 0.3 percent PG (see the bright field). Figure 3 must also have the same labelling if that can be done.
Answer: Thank you for the comments. The labeling for figure 3 was hided because of a formatting issue, we have adjusted it to show the full labels. Due to difference in physical properties, cell confluency in the photos was different. To compensate for this difference, we used SSEA4/DAPI intensity and F-actin/DAPI intensity for quantification.
7. If a concentration of HA that is 3 percent is hazardous to the cell, then why did the authors choose this concentration for their experiment? Could you elaborate?
Answer: Thank you for the comments. Please refer to our answer to question #5 above. 3% HA was shown to be biocompatible by previous research and a popular material for MSC encapsulation for in vivo delivery. In our experimental setting, the cells were encapsulated at high density with almost no fresh media supply, which would allow us to observe the effects of different materials on cells viability and applications.
8. Why did the authors get rid of the 3% HA group in Figure 4? The author should include the data for HA as well for the comparison.
Answer: Thank you for the suggestion. 3% HA is a non-crosslinked material, though it is suitable for injection delivery, it can be easily dissolved if it is immersed in substantial media for biologics release collections. Therefore, we were unable to collect enough conditioned media from 3% HA encapsulated MSC for Mass Spectrometry analysis. However, Hystem is a HA based cross-linkable hydrogel, therefore, data from Hystem are not only representing Hystem itself, but also would be relevant to HA based hydrogels.
9. The assertion made in the section title for 3.3 does not accord with the findings presented in Figure 4.
Answer: Thank you for the comment. Section title for 3.3 is “hADMSCs in PGmatrix secreted high immune responsive proteins and comparable levels of VEGF-A and TGFβ1 factors ”. The “comparable levels” are referring to levels of 3D material groups. To make it clear, we have revised the title as “hADMSCs in PGmatrix secreted high immune responsive proteins and comparable levels of VEGF-A and TGFβ1 factors as other hydrogels”
10. Make sure the labels are correct in Figure 5A.
Answer: Thanks, we double checked the labels in Figure 5A and they are correct.
11. The author is responsible for providing evidence of the materials' in vivo
biocompatibility. In light of the fact that the in vitro results showed high toxicities, a subcutaneously implanted material biocompatibility study can be carried out.
Answer: Thank you for the comment. Please refer to our answer to question #5, biocompatibility of the hydrogels was proved by previous studies. In this study, the fast-decreasing viability was caused mainly by the limited media and high cell density encapsulated.
12. If, on day 21, all of the animals had the same level of recovery, then what difference does it make if the PG Matrix material initially boosted healing?
Answer: Thank you for the comments. We have explained in the discussion ( highlighted as blue). For your convenience, it is copied here: “The healthy outbred mice were chosen with the purpose of better representing a heterogeneous population compared to inbred diabetic strains, however, the diabetes situation induced with STZ in the healthy outbred mice did not provide significant effects of slower healing complexity as expected by day 21 although
all mice exhibited diabetic symptoms before wound creation. Nevertheless, healing differences was obviously observed at earlier timepoint such as day 4 (Figure 5B), and our MS analysis data (Figure 4B) suggested that PGmatrix encapsulation promoted release of immune responsive proteins, which would help reduce inflammation and contribute to faster wound healing as observed in early stage of the healing process. For more complex and larger wounds, the
promotional healing effects of encapsulated hADMSC might be more bservable, as reported in some chronic, nonhealing wounds [59]. In summary, pretreatment of MSC with hydrogel encapsulation may be better suited for use in promoting anti-inflammatory and immunosuppressed environments, and thus suitable for use in inflammatory and autoimmune diseases. PGmatrix encapsulation would be a better alternative to HA hydrogels as it improves maintenance of cell viability. Nonetheless, choosing optimal MSC delivery hydrogels merits further exploration.”
13. The information in Figure 6 suggests that all of the materials are of equal quality. For this reason, I feel that the title of the article would lead the readers astray. Take into consideration altering the title of the manuscript.
Answer: Thank you for the suggestion. Please refer to our answer to question 12. The healing effect might be more obvious if the experiment had stopped at an earlier timepoint. We revised the title as “Injectable Peptide Hydrogel Encapsulation of Mesenchymal Stem Cell Improved Viability, Stemness, Anti-Inflammatory Effects, and Early Stage Wound Healing” though it is a
little longer but reflects the content more accurately.
14. Author might want to consider reworking the part about the conclusion.
Answer: Thank you for the comments. As explained in previous answers, our result showed that PGmatrix help maintained cell viability under harsh condition and stimulated secretion of immune response-related protein, with VEGF-A and TGFB1 levels comparable to other hydrogels. Though the in vivo experiment showed the same wound healing effects by day 21, it did demonstrated significant improved healing at early stage before day 7, which is meaningful
for treating non-healing wounds.
Reviewer 2 Report (New Reviewer)
Thank you so much for the great article! There are a few points that need to be clarified.
1. all figures have text like "Figure 1". There is a caption to the figure, you don't need it additionally.
2. OD and Viability must correlate, that is, the increase or decrease must go the same way in both cases. This is not the case with you. Add the formula for converting from OD to Viability. Viability should be expressed as a histogram, bar graph. Add information about control cell on the histogram graph. There can't be a graph as a line with dots. An OD graph can be a line with dots. Apply discontinuities at long intervals in the graphs for more correct graphs.
3.Figure 2 and beyond-if you're giving a general skail bar in the figure, write that in the caption to the figure. Without that, you need to include a scale bar in each picture.
4. Figure 3 - the caption in the figure is cropped, must be corrected
5. Figure 4 - the color identification of the same patterns should be the same on the whole panel of patterns, fix it.
6. In the text there is a color highlighting. it is not clear why this is needed.
7. Check for inaccuracies in the text and punctuation.
Thank you!
Author Response
Thank you so much for the great article! There are a few points that need to be clarified.
1. all figures have text like "Figure 1". There is a caption to the figure, you don't need it additionally.
Answer: Thank you for the suggestion. We have removed the large “Figure” texts.
2. OD and Viability must correlate, that is, the increase or decrease must go the same way in both cases. This is not the case with you. Add the formula for converting from OD to Viability. Viability should be expressed as a histogram, bar graph. Add information about control cell on the histogram graph. There can't be a graph as a line with dots. An OD graph can be a line with dots. Apply discontinuities at long intervals in the graphs for more correct graphs.
Answer: Thank you for the valuable comments. In our case, interaction between hydrogel and cell will affect cell metabolism, OD reflects the combined effects of viability and metabolic activities, as stated on page 6, line 22 and page 15, line 10. Therefore, the trend for OD and viability should not be exactly correlated, which is why the cell viability was determined separately from the OD experiments. The viability was measured as an independent experimental data using AO/PI staining and cell counting with a cellometer. We have added
viability calculation formula “Viability = live cell number / total cell number” in the Methods section (page 3, line 146). Figure 1A, 1B were changed to bar graphs.
3. Figure 2 and beyond-if you're giving a general skail bar in the figure, write that in the caption to the figure. Without that, you need to include a scale bar in each picture.
Answer: Thank you for the comments. We have revised figure captions to include general scale bar. For Figure 6A, the scale bar in each image was changed to black with larger text with description in the caption.
4. Figure 3 - the caption in the figure is cropped, must be corrected
Answer: Thank you for pointing out. We have adjusted the format to reveal the labels.
5. Figure 4 - the color identification of the same patterns should be the same on the whole panel of patterns, fix it.
Answer: Thank you for the comments. We have changed figure 4E into bar graph and adjusted the color coding for consistency.
6. In the text there is a color highlighting. it is not clear why this is needed.
Answer: Revised texts are marked in blue, those are revision based on previous reviewer opinions. Previous color highlighting has been erased. All blue text in the new version are either revisions or highlighted for answering reviewers’ questions in this second review.
7. Check for inaccuracies in the text and punctuation.
Answer: Thank you for the suggestions. We have revised the manuscript to correct inaccuracies.
Thank you!
Round 2
Reviewer 1 Report (New Reviewer)
If the authors had seen a decrease in the number of viable cells after 24 hours, they should have made an effort to lower the cell concentration. If the observed cell viability on day 0 was 100, and it dropped to 50 in 24 hours, the situation is as follows: It is equivalent to the IC50 level. This will not be counted as an improvement to the viability of the cells. Additionally, a suitable amount of media should be provided. Before proceeding with the following investigation, the author should have made sure that cell viability was maximized.
Even though the authors have conducted in vitro tests to test the survivability of cells in the hydrogels, the results showed that the cells were damaged, which the authors stated was representative of the scenario in the real world. If the cell viability of the treatment is just 50%, then there is a genuine cause for concern, and the treatment needs to be rethought.
It is strongly suggested to conduct an in vivo subcutaneous biocompatibility investigation.
Author Response
Please see attachment

This manuscript is a resubmission of an earlier submission. The following is a list of the peer review reports and author responses from that submission.
Round 1
Reviewer 1 Report
Overview:
In the manuscript entitled “Injectable Peptide Hydrogel Encapsulation of Mesenchymal Stem Cell Improved Viability, Stemness, and Anti-Inflammatory Effects” the authors report a new peptide-based hydrogel (PGmatrix) able to sustain and to improve viability of hADMSC compared to widely used hydrogels in literature.
Unfortunately, there are some concerns:
-The authors assume that (Stage-Specific Embryonic Antigen 4, SSEA4) is also a marker for stemness of hADMSC. Could you give us some reference about that? In order to demonstrate that the hydrogel did not spontaneously induce any typical MSC differentiation thus preserving the hMSC stemness, even when encapsulated, the authors could analyze the gene expression profile for Adipsin and Fatty acid binding-protein 4 (FABP4) identified as markers for adipocytes.
-It is not clear this part :Since PGmatrix does not contain cell adhesion sites, the SSEA4 upregulation effect might relate to mechanical cues provided by the PGmatrix environment. According to actin staining images (Figure 3), hADMSCs had more extreme actin agglomeration when encapsulated in PGmatrix than in HA or Hystem. Researchers have found that when in 3D configuration, MSCs can cluster together to form spheroids. Relaxation of actin tension would improve MSC stemness and therapeutic effects. More complete actin relaxation of hADMSC in PGmatrix may explain the SSEA4 upregulation effect and suggested better stemness of these hADMSCs compared to those in suspension or encapsulated in HA hydrogels.
Authors reported a different actin tension in the encapsulated cells. What does is it exactly mean ? It is really tricky to appreciate these differences in the pictures proposed. My suggestion is to acquire the sample with a higher magnification and quantify in somehow the actin.
The authors declare that The heterogeneity from the outbred mice strain CD-1 may also be 486 the reason causing large statistical variations among the groups. Thus, the mice model used might not be sufficient to investigate the effects of hADMSC delivery with hydrogels 488 for this type of diabetic skin wound . So , why don’t they change the model? or the effect might be quantified by the macrophage infiltration ?
Reviewer 2 Report
In the present study, a new peptide-based hydrogel (PGmatrix) was applied. And data revealed that hADMSCs encapsulated with PGmatrix showed improved healing of skin wound in diabetic-induced mice at early stage, suggesting possible anti-inflammatory effects though similar re-epithelialization and collagen density were observed among PGmatrix and HA or Hystem hydrogels.
Any advantage for skin wounds in diabetics? no data revealed;
How about anti-inflammatory effects in in vivo?
Reviewer 3 Report
This paper describes the analysis of cells cultured in a set of hydrogel scaffolds. The work focuses on a peptide hydrogel and compares it to hyaluronic acid hydrogels, which are more commonly used. The authors conclude that the peptide based hydrogel improves cell viability in vitro and wound closure in vivo. Although there are some bits of promising results there are major issues with the work. The entire focus of this work is that changing the 3D cell culture scaffold will led to better outcomes for cells encapsulated in these scaffolds and, in turn, wound resolution. Although this is the focus, the hydrogel scaffolds are not characterized. No details are provided about the peptide hydrogel at all. Also the 3 scaffolds are made with different percentages of materials, but since no characterization is done it is unclear if these result in completely different environments for the cells. Since the characterization is missing it is impossible to interpret the data but there is data interpretation provided. As a reviewer I cannot assess the validity of the interpretation since the base material is never measured. Due to this and other major comments detailed below, in the current form this manuscript is not suitable for publication. This manuscript might be suitable for publication after a large amount of additional experiments and major revisions are done, but that is still unclear since major pieces of experimental data are missing and additional review would be required.
Major comments, questions and concerns:
- The introduction is very brief. The reason for hydrogel delivery systems for 3D cell culture is motivated but the work is not well placed in the literature in terms of peptide hydrogels. Instead of recognizing this large class of gels the authors only focus on their specific gel. The introduction should introduce peptide gels and the current state of the art of these materials, especially in regards to 3D cell culture and as cell delivery vehicles.
- The introduction also does not provide the types of material properties and cell outcomes that are needed for a material to be a successful delivery scaffold.
- The specific cell being delivered is also not really introduced, the cell type is but hADMSCs are not.
- What is in the PGmatrix? It is important to know this to know what cues are being provided to the cells from the material.
- What are the properties of the gel materials being compared? Are they all the same stiffness? Do they provide the same cues to the encapsulated cells? The environments should be kept consistent between the 3 different materials.
- Were the hydrogels characterized? This seems to be missing from the manuscript and is really important.
- I am very concerned that the hydrogels were not characterized. Since this is not done it is impossible to tell whether it is the chemistry, properties or cues provided by the material that is changing cell behavior and outcomes. Due to this, it is impossible to interpret any data collected in this study.
- For cell viability, it seems that the sample is squeezed out of a syringe for the measurement, is there any consideration to the effect of shear on the cells and how this might change the viability measurements?
- Is there an explanation for the extremely low viability measured only 48 hours after encapsulation? In other gel systems the viability is much higher for much longer.
- It seems long term healing is not changed with any scaffold, this seems to be a negative outcome, can the authors comment on why earlier increased wound closure would be beneficial, since this is not discussed at all.
- The shear thinning behavior of the PG matrix is first mentioned in the discussion, how was this characterized?
- There are many details provided in the discussion which are hard to follow and there is not much interpretation of these details.
- Why were hydrogels kept in syringes and not cultured in plates, which is standard? It seems strange that for an application these materials would be kept in a syringe for such a long period of time.